# Effect of Space Order on Impulse Buying: Moderated by Self-Construal

**DOI:** 10.3390/bs13080638

**Published:** 2023-07-31

**Authors:** Yi Shi, Jaewoo Joo

**Affiliations:** Department of Marketing, College of Business Administration, Kookmin University, Seoul 02707, Republic of Korea; sy897972158@outlook.com

**Keywords:** impulse buying, space order, self-construal, off-line store, sales

## Abstract

Objective: Impulse buying is a recognized phenomenon as consumers have abundant shopping opportunities. We investigate whether orderly space encourages consumers to buy impulsively and whether this relationship is moderated by self-construal. Specifically, we hypothesize that consumers show greater impulse buying intentions when space is orderly than disorderly. We also hypothesize that when interdependent self-construal is primed, the effect of orderly space on consumers’ increased impulse buying intentions will be attenuated. Background: Our hypotheses are based on the research about emotions that consumers experience while they shop in a retail store. When the store is orderly, consumers experience pleasure. In contrast, disorganized shelves, unsorted merchandise, and messy clothing racks evoke negative emotions. A recent study shows consumers’ positive emotional responses to a retail environment result in heightened impulse buying. Methods: Two experiments were carried out to test the two hypotheses. Experiment 1 employed a 2 (space order: orderly vs. disorderly) between-subjects design. Participants randomly received one of the two store images and were asked to indicate their impulse-buying intentions. Experiment 2 employed a 2 (space order: orderly vs. disorderly) × 2 (self-construal: independent vs. interdependent) between-subjects design. Participants were randomly given one of the two store images and one of the two self-construal priming tasks to measure their impulse buying intentions. Results: As hypothesized, Experiment 1 demonstrated that participants exerted stronger impulse-buying intentions in an orderly space. Experiment 2 also showed that when participants were primed by interdependent self-construal, their impulse buying intentions did not differ, regardless of whether the space was orderly. Implications: Our findings provide insights for offline store managers. To nudge visitors to buy impulsively, managers should organize their spaces orderly. However, the effect of space order on consumers’ impulse buying will disappear when consumers’ interdependent self-construal is activated. Our findings contribute to the academic research into the factors that lead consumers to buy impulsively.

## 1. Introduction

Impulse buying is a significant factor in the modern retail industry. Particularly, the COVID-19 crisis has led to a surge in this phenomenon. Recent polls have indicated that American consumers’ average monthly spending on impulse purchases has increased by 18% since the beginning of the COVID-19 pandemic [1]. Despite the pandemic’s detrimental impact on the global economy, experts attribute 20% of retail sales to consumers’ impulse purchases [2]. These statistics highlight the growing importance of impulse buying in the retail industry. It is strong among Chinese college students. According to China’s 2020 census, young people comprise 20% of the population, and college students are essential to the youth group. The average monthly expenditure of college students is 1954 yuan for now, and a significant proportion of their expenditure goes to impulse buying because emotions often dictate their behaviors.

Impulse buying could be caused by consumers’ exposure to stimuli while shopping, as various stimuli within the store, either directly or indirectly influence them. Store elements such as ambience (lighting, scent), visual design (store layout, shelving, displays, store organization, and storage), and social factors (employees and other customers) can all contribute to impulse buying [3]. However, we also acknowledge that consumer characteristics also impact impulse buying. Therefore, this study focuses on the influence of environment and consumer characteristics on impulse buying.

Our intuition is that the emotion evoked by space order determines consumers’ intentions to buy impulsively. Research suggests confused customers react negatively by exhibiting decreased loyalty and trust, postponing or abandoning purchases, and even changing brand preferences [4]. Another study showed that cognitive load and low processing fluency reduce preferences for products as well [5]. Indeed, consumers’ negative perceptions of stores negatively impact several aspects of the perceived value of products, including perceived time and effort costs, psychological costs, service quality, and product quality. In other words, pleasant consumers buy impulsively, whereas overwhelmed and perplexed consumers may lose impulse buying intentions.

## 2. Literature Review

### 2.1. Impulse Buying

Impulse buying is an unplanned and unintended purchase preceded by exposure to a stimulus and a sudden and powerful buying urge, made without much reflection. Therefore, it was interchangeable with unplanned purchases [6,7]. Since impulse buying is not planned but improvisational, it is generally expected to cause a negative consumer evaluation after a purchase. However, some prior studies show that impulse buying results in positive consumer evaluations [8]. Indeed, although purchasing impulsively leads consumers to fail to consider other alternatives or future influences, doing so is not innocuous but acceptable in society [9,10]. 

Although sensory contact with a product often triggers impulse buying temptation, consumers’ positive mood in an environment is also conducive to it [11]. Differently from the visceral state, which leads consumers to buy a specific product impulsively, mood encourages them to buy any product in a given store impulsively [12]. In sum, the temptation can be activated and intensified by emotion driven by situational factors such as environment or individual factors such as personality [13]. 

### 2.2. Space Order 

Space order is the degree to which space as a physical environment is ordered. 

Orderly space evokes positive emotion. One study shows in a traditional retail store context that, when it is ordered, consumers with strong shopping motives experience pleasure [11,14]. Indeed, consumers stay longer in pleasant spaces while leaving unpleasant ones. In contrast, disorganized shelves, unsorted merchandise, and messy clothing racks evoke negative emotions. Although some studies show that a somewhat disorderly space facilitates new product adoption [15], disorganized grocery stores can impact the efficiency of information processing, which detrimentally impacts consumers’ reaction behaviors, enjoyment, and perception of the attractiveness of the environment [16]. People avoid store environments characterized by high confusion and are less likely to make unplanned buys [3]. In sum, disorderly spaces stimulate consumers’ negative emotions and influence their behavior negatively.

Although no research tested the relationship between space order and impulse buying, prior research implies that they are emotion-related. Orderly space evokes positive emotion, which encourages impulse buying, whereas disorderly space evokes negative emotion, suppressing impulse buying. Indeed, one recent study reports that consumers’ positive emotional responses to a retail environment result in their heightened impulse buying [17]. Thus, the following hypothesis is proposed:

**H1:** 
*Consumers in an orderly space have stronger impulse buying intentions than those in a disorderly space.*


### 2.3. Self Construal

Self-construal reflects the extent to which individuals view themselves either as an individual entity or concerning others. It is influenced by self-knowledge and self-concept, which shape how individuals perceive themselves in relation to others [18]. Self-construal is divided into independent and interdependent [19]. The people with an independent self-construal are centred on the “me” and value their uniqueness and self-discipline. In contrast, the people with an interdependent self-construal are “we” centric people who value maintaining and coordinating relationships within a group. Cultural differences between individualistic and collectivistic societies correspond to these two types of self-construal [18]. Research shows that independent self-construal (vs. interdependent self-construal) positively correlates with impulse buying behavior in consumers from various countries, including Australia, the United States, Hong Kong, Singapore, and Malaysia [20].

Understanding cultural contexts such as individualism and collectivism is essential for comprehending impulse buying behavior [20]. Studying cultural aspects can further assist academics and practitioners in better understanding the impulse buying phenomenon. More independent individuals will engage in greater impulse buying behavior than those whose self-concepts are interdependent. Similarly, collectivist consumers engage in less impulse purchase behavior than individualist consumers. Because impulse consumption is often considered an unplanned and immature behavior that may reflect badly on the group in interdependent societies, people with an interdependent self-construal may be more likely to activate self-regulation goals and thus suppress impulse urges than those with an independent self-construal. Conversely, people with an independent self-construal may be more likely to activate pleasure-seeking goals and, thus, act in a manner consistently [21,22].

Therefore, it may be more probable that individuals with an independent self-construal may ignore the negative effects of impulse buying and exhibit this behavior. Conversely, those with an interdependent self-construal are more aware of the negative impacts of impulse spending behavior and thus refrain from it. Therefore, we propose the following hypotheses (see Figure 1):

**H2a:** 
*Consumers with independent self-construal will have more impulse buying intentions in an orderly than a disorderly space.*


**H2b:** 
*Consumers with interdependent self-construal show no difference in impulse buying intentions in orderly and disorderly spaces.*


## 3. Studies

### 3.1. Experiment 1

Experiment 1 comprised a pre-test and a main experiment. The pretest was conducted to validate the manipulations of space order used in the main experiment, and the main experiment aimed to investigate hypothesis H1.

#### 3.1.1. Purpose

The purpose of Experiment 1 was to investigate the impact of space order on impulse buying behavior. To achieve this objective, the study first manipulated the space order and then measured the participants’ intention to engage in impulse buying.

#### 3.1.2. Stimuli

This study focused on clothing purchases because apparel products are sensory-experience items with symbolic implications frequently purchased based on consumers’ emotional preferences [23]. In line with the experimental manipulation of space order tested in a previous study [24], images showing the clothing products displayed in an actual store were selected as the stimuli for this experiment. For example, in the orderly condition image, the clothes are neatly arranged. In the disorderly condition image, the clothes are scattered. Adobe Photoshop software package was used to produce the images.

#### 3.1.3. Pre-Test 

To determine the suitability of the experiment’s stimuli, we conducted a pre-test and verified the credibility of manipulating space order. In the pre-test, 20 college students were randomly given one of the two images, an orderly store image or a disorderly store image adapted from the previous study [24] (M_age_ = 21.300, SD_age_ = 1.129, Female = 60%). The image provides a walkthrough of the store and its displayed items. 

Next, participants answered two questions about how they felt about the store in the images. These two manipulation check questions were (1) to what extent you think this store is orderly and (2) how messy you think this space is (1 = not at all, 7 = very messy) [25]. The second item was reverse-coded to create the space order index, and the two items’ scores were then averaged. The pre-test results showed that participants perceived the orderly store as more orderly than the disorderly store (M_orderly_ = 5.700, SD_orderly_ = 1.033 vs. M_disorderly_ = 2.150, SD_disorderly_ = 0.747, t(18) = 8.806, *p* < 0.001), confirming that our manipulation worked as intended. 

#### 3.1.4. Experiment 1 Design

Experiment 1 tested whether an orderly space would increase consumers’ impulse buying intentions (H1). The experiment employed a 2 (space order: orderly vs. disorderly) between-subjects design. The studied population was made up of Chinese university students. Participants were informed that this was a consumer behavior survey used for data collection through wjx.cn accessed on 11 January 2023.

#### 3.1.5. Experiment 1 Procedures and Measures

Participants were instructed to read a shopping scenario related to an image that manipulated space order. After reading the scenario and seeing the image, participants were asked to complete a questionnaire about impulse buying intention on a 7-point Likert scale (1 = not at all, 7 = very much). Three questions include: (1) while I was looking at the product, I felt the urge to buy it, (2) if I saw something that interested me in this store, I would buy it without considering the consequences, and (3) I would buy things in this store even though they were not on my shopping list [26,27]. We checked the manipulation of space order once again by asking the two questions (1 = not at all, 7 = very messy): (1) to what extent do you think this store is orderly, and (2) how messy do you think this space is? Finally, gender and age were asked at the end of the experiment.

#### 3.1.6. Experiment 1 Results and Discussion

##### Participants

Experiment 1 was conducted online over two weeks in 2023. 86 Chinese college students participated in the survey, with 80 valid surveys used in the final analysis. Demographic analysis of the sample indicated that 44% of the participants were male and 56% were female. The mean age of the participants was 20.363 (SD_age_ = 1.664), with the youngest participant being 18 years old and the oldest being 24 years old. 

##### Manipulation Checks

To check the manipulation of space order, a statistical analysis was conducted by calculating the mean scores of the two experimental groups using an independent samples *t*-test. Similar to the findings obtained from the pre-test, participants perceived the orderly store to be more orderly than the disorderly store, as evidenced by the mean scores of the two conditions (α = 0.870; M_orderly_ = 4.950, SD_orderly_ = 1.097 vs. M_disorderly_ = 2.675, SD_disorderly_ = 1.089, t(78) = 9.309, *p* < 0.001). These findings suggest that manipulating space order was successful, and the participants perceived the two conditions as distinctly different.

##### Hypothesis Testing

To test whether the impulse buying intentions were influenced by the space order, independent *t*-tests on the impulse buying intentions of the two groups were performed separately. The results revealed that participants in the orderly space condition showed higher impulse buying intentions than in the disorderly space (α = 0.849; M_orderly_ = 4.242, SD_orderly_ = 1.091 vs. M_disorderly_ = 2.825, SD_disorderly_ = 0.887; t(78) = 6.372, *p* < 0.001), supporting H1 (see Figure 2).

### 3.2. Experiment 2

Similar to Experiment 1, Experiment 2 comprised a pre-test and a main experiment. The pretest was conducted to validate the manipulations of self-construal used in the main experiment, and the main experiment aimed to investigate hypothesis H2a and H2b.

#### 3.2.1. Purpose

The purpose of Experiment 2 was to investigate the impact of space order on impulse buying behavior moderated by self-construal. The study first manipulated space order and self-construal to achieve this objective and then measured the participants’ impulse buying intentions. 

#### 3.2.2. Stimuli

According to a previous study [28], when people read a brief description of “a trip to the city” and were instructed to circle all pronouns in the text, independent self-construal is activated if pronouns are independent and interdependent self-construal is activated if pronouns are interdependent. Note that the two versions of a brief description differed only concerning whether the pronouns are independent (e.g., I, me, mine) or interdependent (e.g., we, us, ours). The two versions are below. 

Independent self-construal: I go to the city often. My anticipation fills me as I see the skyscrapers come into view. I allow myself to explore every corner, never letting an attraction escape me. My voice fills the air and streets. I see all the sights; I window shop, and everywhere I go, I see my reflection looking back at me in the glass of a hundred windows. At nightfall, I linger; my time in the city is almost over. When I finally must leave, I do so knowing I will soon return. The city belongs to me.

Interdependent self-construal: We go to the city often. Our anticipation fills us as we see the skyscrapers come into view. We allow ourselves to explore every corner, never letting an attraction escape us. Our voice fills the air and street. We see all the sights, we window shop, and everywhere we go, we see our reflection looking back at us in the glass of a hundred windows. At nightfall, we linger, our time in the city almost over. When finally, we must leave, we do so knowing that we will soon return. The city belongs to us.

#### 3.2.3. Pre-Test 

We tested whether priming shifts the balance between independent and interdependent self-construal. To verify the priming task, we pre-tested 20 college students randomly recruited in China (M_age_ = 21.400, SD_age_ = 1.536, Female = 55%). They were randomly divided into two groups and asked to circle all pronouns while reading one of the two descriptions. 

After circling pronouns while reading the description, they answered six questions about self-construal adapted from a previous study on a 7-point Likert scale (1 = strongly disagree, 7 = strongly agree) [26]. The three questions are about independent self-construal: (1) I thought about myself while reading this passage, (2) I think this passage is all about me, and (3) As I read this, my mind focused on myself. The other three questions are about interdependent self-construal: (4) I thought of myself and my friend while reading this passage, (5) I think the writing is centered on my friend and me, and (6) As I read this, my thoughts focused on my friend and me. Their responses to the first three items were averaged to form an independent self-construal score, and their responses to the other three items were averaged to form an interdependent self-construal score. 

Their collected responses showed that priming successfully manipulated self-construal. Participants who circled independent pronouns (I, me, mine) thought more about themselves than their friends (M_independent_ = 4.367, SD_independent_ = 0.637 vs. M_interdependent_ = 2.867, SD_interdependent_ = 0.849; t(18) = 4.468, *p* < 0.001). In contrast, participants who circled interdependent pronouns (we, us, ours) thought about themselves and friends equally (M_independent_ = 3.633, SD_independent_ = 0.331 vs. M_interdependent_ = 4.233, SD_interdependent_ = 0.817; t(18) < 1). 

#### 3.2.4. Experiment 2 Design

The objectives of Experiment 2 were, firstly, to replicate the findings of Experiment 1, namely, whether the space order increases the impulse buying intention (H1) and, secondly, whether the self-construal of consumers moderates this effect (H2a, H2b). This experiment employed a 2 (space order: orderly vs. disorderly) × 2 (self-construal: independent vs. interdependent) between-subjects design to achieve these goals. Similar to Experiment 1, the questionnaire was designed, and the answers were collected using the China Survey Website accessed on 14 April 2023 (https://www.wjx.cn/). The stimuli in Experiment 2 were identical to those utilized in Experiment 1. However, unlike Experiment 1, the participants in Experiment 2 first performed a priming task before being exposed to a store image. The subjects’ self-interpretation was primed using the methods proposed in previous research [28]. We predicted that when participants primed independent self-construal, they tended to have higher impulse buying intentions. Correspondingly, participants primed with interdependent self-construal might be more likely to activate self-regulation goals, thus suppressing impulse buying intentions. 

#### 3.2.5. Experiment 2 Procedures and Measures

Like Experiment 1, participants indicated their impulse buying intentions by answering the three questions. They include: (1) while I was looking at the product, I felt the urge to buy it, (2) if I saw something that interested me in this store, I would buy it without considering the consequences. And (3) I would buy things in this store even though they were not on my shopping list [26,28] (1 = not at all, 7 = very much). They also answered gender and age.

#### 3.2.6. Experiment 2 Results and Discussion

##### Participants

Experiment 2 was conducted online over a month in 2023. 181 Chinese college students participated, and 167 valid surveys were used in the final analysis. A demographic analysis of the sample indicated that 46% of the participants were male and 54% were female. The mean age of the participants was 20.623 (SD_age_ = 1.531), with the youngest participant being 18 years old and the oldest being 24 years old. Participants were randomly assigned to one of four groups.

##### Manipulation Checks

We checked two manipulations. First, the manipulation of space order was successful again (α = 0.733; M_orderly_ = 4.768, SD_orderly_ = 1.129 vs. M_disorderly_ = 2.681, SD_disorderly_ = 0.875, t(165) = 13.344, *p* < 0.001). Second, participants primed by independent self-construal (α = 0.727; M_independent_ = 4.372, SD_independent_ = 0.691) thought more about “me” compared to participants primed by interdependent self-construal (α = 0.901; M_interdependent_ = 2.895, SD_interdependent_ = 0.880; t(82) = 8.576, *p* < 0.001). In contrast, participants primed by interdependent self-construal (M_interdependent_ = 4.881, SD_interdependent_ = 0.785) thought more about “we” compared to participants primed by independent self-construal (M_independent_ = 3.244, SD_independent_ = 0.522; t(81) = −11.162, *p* < 0.001), suggesting that self-construal was manipulated as planned. 

##### Hypothesis Testing

We analyzed variance (ANOVA) to test whether impulse buying intentions were influenced by space order and self-construal. We obtained findings that the interaction effect of space order and self-construal was significant (F(3, 90) = 24.495, *p* < 0.001).

First, H1 was confirmed again, as participants had higher impulse buying intentions in an orderly space than in a disorderly space (α = 0.896; M_orderly_ = 3.655, SD_orderly_ = 1.290 vs. M_disorderly_ = 2.777, SD_disorderly_ = 0.883, t(165) = 5.119, *p* < 0.001). Second, when participants were primed by independent self-construal, they had higher impulse buying intentions in an orderly space than in a disorderly space (M_orderly_ = 4.349, SD_orderly_ = 1.049 vs. M_disorderly_ = 2.894, SD_disorderly_ = 0.911, t(82) = 6.771, *p* < 0.001). However, when participants were primed by interdependent self-construal, impulse buying intentions did not differ between the orderly space and in the disorderly space (M_orderly_ = 2.659, SD_orderly_ = 0.848 vs. M_disorderly_ = 2.944, SD_disorderly_ = 1.121, t(81) = 1.308, *p* = 0.195). Therefore, H2a and H2b was supported. 

Note that, only in the orderly space, participants primed by independent self-construal had higher impulse buying intentions than participants primed by interdependent self-construal (M_independent_ = 4.349, SD_independent_ = 1.049 vs. M_interdependent_ = 2.944, SD_interdependent_ = 1.121, t(83) = 5.965, *p* < 0.001). Impulse buying intentions in the disorderly space did not differ regardless of self-construal (see Figure 3).

## 4. Conclusions

### 4.1. Discussion

This study used two experiments to test whether space order and self-construal influence impulse buying intention. We obtained findings from Experiment 1 that participants indicated greater impulse buying intentions when the space was orderly than when it was disorderly (H1). Experiment 2 revealed that when participants were primed by independent self-construal, they had higher impulse buying intentions in order than disorderly (H2a). However, when participants were primed by interdependent self-construal, there was no difference in their impulse buying intentions in either the orderly or the disorderly space (H2b). 

Our experimental findings clearly showed that, in the orderly space, there is a statistical difference in impulse buying intentions between independent and interdependent self-construal participants. Specifically, consumers have stronger impulse buying intentions in an orderly space only when they activate independent construal. When interdependent self-construal activates, they inhibit impulse buying intentions.

Note that the results of this study are in line with the conclusions of previous studies. Independent self-construal individuals may be more likely to activate hedonic goals and thus act in ways consistent with hedonic goals, such as impulse buying. However, people with interdependent self-construal are more likely to activate self-regulatory goals and thus suppress impulse buying intentions [20,22]. In collectivist societies, where the emphasis is placed on self-control and emotional moderation, consumers are more inclined to suppress the emotional component of their impulse purchase experience [19]. 

This study has practical significance. It aims to provide insights for offline store managers who strive to increase revenue by providing a comfortable shopping environment or activating consumers with independent self-construal. To nudge visitors to buy impulsively, managers should organize their spaces orderly. However, the effect of space order on consumers’ impulse buying will disappear when consumers’ interdependent self-construal is activated. Our findings also contribute to the academic researchers’ interest in the effect of space order [15] and the role of emotion [12].

### 4.2. Limitations and Future Research Directions

There are several limitations to this study, and overcoming these limitations will contribute to future research. First, since this study is based on Chinese college students, we attempt to understand consumers’ impulse buying intentions based solely on an Eastern point of view. In the East, the collectivist notion of self emphasizes interdependence and emotional control. However, in the West, the individual notion of self emphasizes individual wants and desires and hedonistic enjoyment [19]. Proper consideration of these differences is crucial; therefore, it is necessary to conduct further research on more diverse groups, leading to more extensible conclusions in theory and practice.

Second, deep consideration is needed to understand marketing strategies that promote impulse buying. Looking at how different variables encourage impulse buying, and which have the most impact will be helpful, particularly in diverse cultural contexts. In Asia, for instance, studies report that the messages designed by firmly established behavioral theories such as curiosity, endowment, or bandwagon did not increase responses, or widely popular package designs such as changing cuteness or color fail to draw consumers’ responses [27,29,30]. Global online markets make it increasingly necessary to study processes that may affect people from other countries or regions differently. A future study could investigate the interaction of situational variables within different cultural settings and among consumers with different levels of impulsiveness.

Finally, this study was conducted in a fictitious online environment, and many factors affected the results of the experiment, such as the environment and the temporary mood of participants at that time. Therefore, we acknowledge this limitation and suggest that future research should be conducted in real-life situations to understand impulse buying behavior [29] better.

## Figures and Tables

**Figure 1 behavsci-13-00638-f001:**
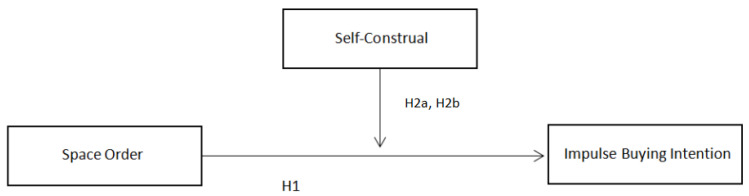
Research framework.

**Figure 2 behavsci-13-00638-f002:**
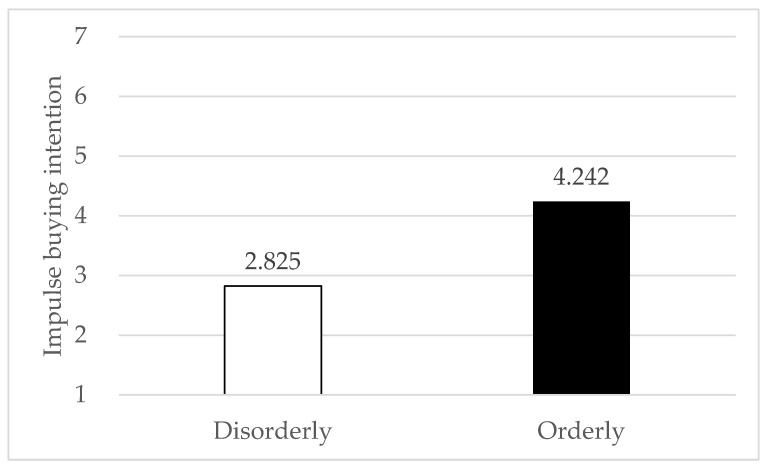
Impulse buying intention as a function of space order.

**Figure 3 behavsci-13-00638-f003:**
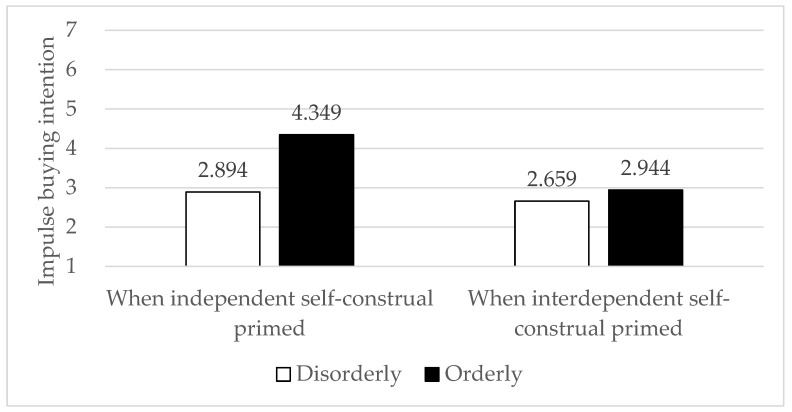
Impulse buying intention as a function of space order and self-construal.

## Data Availability

This study’s data collected and presented are available upon request.

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
