# Peer review of "Effect of Space Order on Impulse Buying: Moderated by Self-Construal"

_behavsci, 2023, doi:10.3390/bs13080638_

Round 1

Reviewer 1 Report

1. The main references in this paper: The influence of culture on consumer impulse buying behavior (Kacen & Lee 2002) is for four countries, two individualist countries (Australia and the United States) and two collectivist countries (Singapore and Malaysia) 706 students and non-students were surveyed. Compared with the number of samples in this paper, the number of samples is obviously low and there is no diversity.

2.  In China, the questionnaire was designed through the China Survey Network (wjx.cn), and the answers collected are not convincing and lack reliability and validity.

There are still grammatical errors and inappropriate text descriptions in English.

Author Response

Thank you very much for your review, please see the attachment.

Response to Reviewer 1 Comments

Point 1. The main references in this paper: The influence of culture on consumer impulse buying behavior (Kacen & Lee 2002) is for four countries, two individualist countries (Australia and the United States) and two collectivist countries (Singapore and Malaysia) 706 students and non-students were surveyed. Compared with the number of samples in this paper, the number of samples is obviously low and there is no diversity.

.

Response 1: Thank you for your suggestion. We will make a more full investigation in the future when conditions permit, and according to your suggestion, we will conduct research on more groups in future research to expand the research results.

Point 2: In China, the questionnaire was designed through the China Survey Network (wjx.cn), and the answers collected are not convincing and lack reliability and validity

Response 2: Thanks for your careful review, we have added Cronbach's alphas in the data section of the text

Reviewer 2 Report

It was a pleasure to read this cleanly executed and well-written paper, and I was very impressed that it was based on a Master’s thesis. The paper uses 2 online surveys of Chinese students to show that intentions to shop impulsively are higher in a tidier clothing store, compared with an untidy (disordered) clothing store. This effect of orderliness was moderated by self-construal. When Chinese students were manipulated to have an interdependent self-construal, as opposed to an independent self-construal, intentions to impulse shop remained low (less than 4.5 on a 7-point scale, i.e., very low probability) no matter whether the store was tidy or disorderly. I have some comments I hope the authors will find useful when revising the paper.

First, it would be good to explain why students were sampled, when it appears that the Chinese version of Amazon’s Mechanical Turk was used, and so a sample representative of the online population, if not the general population, could have been sampled. The introduction argues that “College students are transitioning from adolescence to adulthood, and their actions can be controlled by their emotions, making it easy to engage in impulse buying” (P2 L49). However, this statement is not based on cited evidence, and so there was an opportunity in this study to compare students with other demographic groups (e.g., non-students the same age). Perhaps the reason was an ethical one, because students are only allowed to survey other students?

Second, the results should report standard deviations and effect sizes, so that readers can confirm the results, understand how large the effects were, and future researchers can include these results in meta-analysis more easily. The results should also report the correlation between the 2 items in the orderly scale (after reverse-scoring the second item), and Cronbach’s alphas for the 3-item impulse intentions scale, and the 3-item independent and interdependent scales.

I wish the authors the best of luck with their future research.

There are some minor problems with the writing that can be easily fixed. (1) H2b proposes to test a null hypothesis of no difference. It is possible to test hypotheses of equivalence (Weber & Popova, 2012, https://doi.org/10.1080/19312458.2012.703834), but this requires specifying the effect size and collecting a big enough sample to detect that effect. It would be easier to make H2 one hypothesis about an interaction (moderation) effect: “H2: Self-construal will moderate the effect of an orderly versus a disorderly space on impulse buying intentions, such that this effect will be more significant for consumers with an independent self-construal, compared to consumers with an interdependent self-construal.” The evidence for this hypothesis would be a significant 2-way interaction between space-order and self-construal, which when probed by follow-up t-tests, would show more significance for the independent group compared with the interdependent group. (2) The writing uses several words to mean “responding” to the survey items, but some of these words are misleading. I think all the measures used rating scales, so replace “checked” (P8 L319) and “ranked” (P9 L324) with “rated” at the appropriate places. (3) Do not use the word “prove” (P10 L367) in scientific writing, as in science, nothing is ever proven in the sense of a deductive proof. Science is inductive: we can only report evidence showing support for a hypothesis or rejecting a hypothesis. (4) I think you mean “practical” rather than “academic” in “This study has academic significance” (P10 L393). (5) Is it possible to show the pictures of the stores used in the experiments?

Author Response

Thank you very much for your review, please see the attachment

Response to Reviewer 2 Comments

Point 1: It would be good to explain why students were sampled, when it appears that the Chinese version of Amazon’s Mechanical Turk was used, and so a sample representative of the online population, if not the general population, could have been sampled. The introduction argues that “College students are transitioning from adolescence to adulthood, and their actions can be controlled by their emotions, making it easy to engage in impulse buying” (P2 L49). However, this statement is not based on cited evidence, and so there was an opportunity in this study to compare students with other demographic groups (e.g., non-students the same age). Perhaps the reason was an ethical one, because students are only allowed to survey other students?

.

Response 1: I couldn't agree more with you. Papers need to be rigorous,so I cut out my own subjective speculation ("making it easy to engage in impulse buying"). And according to your suggestion, we will conduct research on more groups in future research to expand the research results.

Point 2: The results should report standard deviations and effect sizes, so that readers can confirm the results, understand how large the effects were, and future researchers can include these results in meta-analysis more easily. The results should also report the correlation between the 2 items in the orderly scale (after reverse-scoring the second item), and Cronbach’s alphas for the 3-item impulse intentions scale, and the 3-item independent and interdependent scales.

Response 2: Thank you for your advice, I have added relevant content in the article.

Point 3: H2b proposes to test a null hypothesis of no difference. It is possible to test hypotheses of equivalence (Weber & Popova, 2012, https://doi.org/10.1080/19312458.2012.703834), but this requires specifying the effect size and collecting a big enough sample to detect that effect. It would be easier to make H2 one hypothesis about an interaction (moderation) effect: “H2: Self-construal will moderate the effect of an orderly versus a disorderly space on impulse buying intentions, such that this effect will be more significant for consumers with an independent self-construal, compared to consumers with an interdependent self-construal.” The evidence for this hypothesis would be a significant 2-way interaction between space-order and self-construal, which when probed by follow-up t-tests, would show more significance for the independent group compared with the interdependent group.

Response 3: Thank you for your suggestion, we will take this part into consideration.

Point 4: The writing uses several words to mean “responding” to the survey items, but some of these words are misleading. I think all the measures used rating scales, so replace “checked” (P8 L319) and “ranked” (P9 L324) with “rated” at the appropriate places.

Response 4: Thanks for your careful review, I have replaced the words according to the suggestion.

Point 5: Do not use the word “prove” (P10 L367) in scientific writing, as in science, nothing is ever proven in the sense of a deductive proof. Science is inductive: we can only report evidence showing support for a hypothesis or rejecting a hypothesis.

Response 5: The words were replaced according to the suggestion.

Point 6: I think you mean “practical” rather than “academic” in “This study has academic significance”

Response 6: Thanks again for your careful review, and the words were replaced according to the suggestion.

Point 7: Is it possible to show the pictures of the stores used in the experiments?

Response 7: I am very sorry that the picture I used was not authorized by the copyright owner, so I can't show the picture in my article. Instead, I can show the pictures here to help you understand the experiment.

Round 2

Reviewer 1 Report

no comment.

no comment.